# History, Current Techniques, and Future Prospects of Surgery to the Sellar and Parasellar Region

**DOI:** 10.3390/cancers15112896

**Published:** 2023-05-24

**Authors:** Cameron A. Rawanduzy, William T. Couldwell

**Affiliations:** Department of Neurosurgery, Clinical Neurosciences Center, University of Utah, 175 N. Medical Drive East, Salt Lake City, UT 84132, USA

**Keywords:** sellar, parasellar, transsphenoidal surgery, transcranial surgery, pterional, endoscopic

## Abstract

**Simple Summary:**

This paper provides a definition of the sella, a description of common pathologies affecting the region, and an overview of the historical development of surgical approaches, as well as the advantages of techniques that are commonly used today.

**Abstract:**

The sellar and parasellar region can be affected by diverse pathologies. The deep-seated location and surrounding critical neurovascular structures make treatment challenging; there is no singular, optimal approach for management. The history and development of transcranial and transsphenoidal approaches by pioneers in skull base surgery were largely aimed at treating pituitary adenomas, which are the most common lesions of the sella. This review explores the history of sellar surgery, the most commonly used approaches today, and future considerations for surgery of the sellar/parasellar region.

## 1. Introduction

The sellar/parasellar region can be affected by a multitude of neoplastic, vascular, and inflammatory pathologies [1]. Its deep-seated location is in close proximity to vital neurovascular structures, and tumor mass effect results in characteristic symptoms including ophthalmoplegia, visual field deficits, cranial neuropathies, and endocrine dysfunction [1]. The surrounding optic chiasm, pituitary–hypothalamic tract, and carotid artery segments make treatment challenging, and even small lesions can cause significant morbidity [2]. In the modern era, transcranial and transsphenoidal approaches to the sellar and parasellar region, with endoscopic or microscopic techniques, are regularly used to treat a diverse collection of pathologies; both possess advantages and limitations when applied in specific cases [3]. The refinement and development of the operative approaches used today represent the progressive evolution of concepts introduced just a century ago [4,5]. In this review, we provide a historical commentary about the pioneers of the development of transcranial and transsphenoidal surgical approaches to the sella, discuss the commonly used, modified, and nuanced procedures of the present, and consider future directions.

## 2. Anatomy

The sella turcica is a bony depression situated deeply in the middle fossa between the anterior clinoid and posterior clinoid processes [6]. Because of the surrounding critical structures and the frequency of substantial anatomic variations seen in the sella and the sphenoid sinus, management of pathology in this location is exceedingly difficult [7]. The pituitary gland sits in the sella turcica, with the optic chiasm positioned superiorly, the cavernous sinus and carotid arteries bounded laterally, the basilar artery and brainstem posteriorly, and the clivus inferiorly [3,8]. The space surrounding the sella turcica is termed “parasellar” space. The suprasellar region is above the sellar diaphragm and extends to the floor of the third ventricle [6]. These sellar and sellar-adjacent regions can be affected by diverse pathologies. The most common lesion of the sella is a nonfunctioning pituitary adenoma, followed by meningiomas (Figure 1) [6,9,10,11]. Meningiomas can arise from the diaphragm, tuberculum, anterior clinoid process, as well as from the chiasmatic sulcus dura and limbus and planum sphenoidale [6,12]. Tuberculum sellae meningiomas comprise nearly 5–10% of all intracranial meningiomas and regularly cause visual disturbances because of local mass effect on the optic apparatus [13,14]. Other pathologies of interest include craniopharyngiomas, Rathke cleft cysts, chordomas, metastases, and aneurysms [1,15]. Because pituitary adenomas make up nearly 90% of all sellar lesions [8], surgery to the region was largely developed to manage these tumors.

## 3. History

The earliest accounts of a transsphenoidal approach can be credited to the ancient Egyptians [16] (Figure 2). During the mummification process, they used transnasal routes to remove the brain with long, customized instruments [17]. Their ancient practices provide insight into an advanced knowledge of anatomy and recognition of transnasal routes to the cerebrum. The origin of neurosurgical approaches to the sella dates to the late 19th century, when Richard Caton and F.T. Paul unsuccessfully attempted a pituitary tumor resection using a temporal approach [18]. Subsequently, transcranial and transsphenoidal approaches were developed to reach the pituitary. From the early days of extensive, disfiguring pituitary surgery with illumination by candlelight in the pre-antibiotic era to the introduction of the operative microscope and then endoscope to facilitate minimally invasive approaches [15,19], the evolution of surgical approaches to the sella was achieved through mentorship, trial and error, collaboration, and innovation.

Caton and Paul attempted their temporal approach to the pituitary at the suggestion of Sir Victor Horsley, who himself was studying the ability to reach the sella turcica via a transcranial–transfacial route [15]. Between 1904 and 1906, Horsley attempted a transfrontal–transtemporal approach to treat pituitary tumors in a case series of 10 patients; he reported a mortality rate of 20% [1,15,18,19]. Around the same time in Austria, Hermann Schloffer was studying the work of Davide Giordano, who proposed the removal a pituitary tumor transfacially using a transglabellar nasal approach [18]. In 1907, Schloffer reported the first successful tumor resection via a superior transsphenoidal approach [15,20,21,22]. Although the success of these early approaches was limited by infection, meningitis, extensive operating time, and cosmetic defects, the interest in transsphenoidal pituitary surgery continued to grow. In 1909, Emil Theodor Kocher implemented the transsphenoidal approach using a submucosal dissection of the nasal septum, which expanded visualization of the sella, reduced risk of infection, and improved cosmetic outcomes [15,18,21]. The principle of submucosal dissection of the septum was a milestone for inferior extracranial approaches [19]. Two Viennese rhinologists, Oskar Hirsch and Markus Hajek, continued to develop transsphenoidal surgery and implemented the lateral endonasal approach through a direct transethmoidal route without reflection of the nose [1,15,18,19,23]. Throughout his career, Hirsch performed the endonasal transeptal transsphenoidal approach on 413 patients, and between 1919 and 1937, reported a mortality rate of only 5.4% [3]. His contributions to transsphenoidal surgery were great, and he continued to develop his practice, with improved outcomes at a time when transsphenoidal surgery was largely left to the wayside.

In 1910, Albert E. Halstead introduced a modified sublabial gingival incision to expose the sphenoid sinus [15]. This approach, as well as previous work by Schloffer, influenced Harvey Cushing and became the impetus for his own contribution to transsphenoidal surgery. Cushing attempted transsphenoidal resection to the pituitary using the approach from Halstead that he modified and refined, and from 1910 to 1925, Cushing treated patients while achieving a 5.6% mortality rate in this pre-antibiotic era [15,19]. Despite the acceptable results of his series, early transsphenoidal surgery was limited by poor visualization and limited resection capability, and Cushing was partial to mastering transcranial techniques. Specifically, he used a transfrontal craniotomy with a direct subfrontal midline approach to treat sellar lesions (Figure 3) [1,18,24]. Nearing the twilight of his career, from 1929 to 1931, Cushing virtually abandoned the transsphenoidal approach in favor of transcranial approaches [15,18,21]. His influence over the neurosurgical community at the time led to an overall decline in transsphenoidal surgery in the 1930s and onwards [15,19,21].

The survival, and eventual revival, of transsphenoidal surgery is largely credited to the work of Norman Dott, Gerard Guiot, and Jules Hardy. Dott, as a fellow from Edinburgh, observed the transsphenoidal approach to the pituitary from Cushing and adopted it into his own practice [18]. In a series of 120 patients, using the transsphenoidal approach, Dott recorded great success and eventually achieved a mortality rate of 0% in his final 80 patients [21]. His abundant experience in pediatric surgery may have contributed to his comfort operating in the dark, narrow corridors inherent in the transsphenoidal approaches of the time [4]. His interest in engineering led to his designing of a speculum with a light attachment to improve illumination and visualization [4,18,19,21]. Guiot, a French neurosurgeon, learned the transsphenoidal technique from Dott, and in the 1950s, he introduced intraoperative radiofluoroscopy to transsphenoidal surgery to achieve visual confirmation of the depth and position of his instruments [4,15,18,19]. In his own series of 1000 patients, Guiot demonstrated superiority of the transsphenoidal approach to the transcranial approach through reduction in morbidity and better visual outcomes for patients with suprasellar tumors [21]. He went on to report the first primitive endoscopic endonasal transsphenoidal approach in 1963, although the quality of endoscopes was poor at the time [15,17]. Finally, Hardy was a mentee of Guiot who revolutionized the transsphenoidal approach. Hardy introduced the binocular microscope to transsphenoidal pituitary surgery in 1967 to overcome many of the disadvantages of the approach at the time [15,19,21]. With the operative microscope, Hardy introduced the concept of selective microadenomectomy to remove the tumor, spare the normal pituitary gland, and preserve endocrine function, and this obviated the need for regular hormonal replacement therapy [18,19,20,21]. The collaboration and mentorship among early innovators led to the development, and survival, of transcranial and transsphenoidal approaches to the sella that became the foundation for techniques used in the present. Today, transsphenoidal surgery is the gold standard for lesions of the pituitary gland and parasellar lesions [7]. Further advancements in intraoperative imaging with endoscope and microscope technology continually expand the opportunities and possibilities in skull base surgery.

## 4. Current Treatment Paradigm

The two main approaches to the sella are transcranial and transsphenoidal [25] (Figure 4). Transcranial surgery can be bi- or unilateral; commonly, a pterional, frontolateral, or orbitozygomatic approach is used. Within the last decade, keyhole approaches have risen in popularity [26]. Bilateral approaches include bifrontal and subfrontal craniotomies [13,14]. Transcranial approaches are typically performed in anterior or anterolateral corridors and are preferred for large tumors, those with extrasellar extension, and fibrous or indurated tumors [1]. They offer a wide exposure, but this comes at the cost of increased brain retraction and/or bone removal for a craniotomy [27,28]. The decision to approach a tumor via open approach versus endoscopically is based on numerous factors including surgeon preference; often, both methods can be viable options for the same skull base lesion [29].

Common approaches include pterional, orbitozygomatic, frontolateral, and more recently supraorbital keyhole approaches (Figure 5; Table 1) [13]. More minimally invasive approaches, including microscopic and endoscopic transsphenoidal surgery, exist to establish direct routes to the sella without the intrinsic morbidity associated with a transfacial or transcranial craniotomy [30,31]. There are advantages and disadvantages to both, and there is extensive overlap between the applicability of each approach. Operative decision-making is contingent on a range of factors including lesion type, growth trajectory and anatomical location, the symptoms at presentation, and the need for vascular control, among others [1]. A proposed grading system by McDermott and colleagues incorporates three key characteristics into determination of the optimal approach: tumor size, optic canal invasion, and arterial encasement [2,32]. Tumors with lower scores, i.e., those with smaller size and no arterial encasement or optic canal invasion were amenable to transsphenoidal surgery, whereas larger tumors with arterial and surrounding structure extension may be better managed with an open approach [2]. Another important tumor characteristic to consider when planning the ideal approach is consistency. Fibrous tumors are less amenable to suction and dissection and can present with adhesions; these tumors are better managed with an open approach [33].

Generally, the open craniotomy is the gold standard for meningiomas, such as tuberculum sellae meningiomas, because of the access to and ability to visualize the optic chiasm and nerves [33]. In contrast, in the case of pituitary adenomas, neurosurgeons are moving toward regular use of an endoscopic approach [34]. The endoscopic endonasal approach for tuberculum sellae meningiomas is most appropriate when the tumors are small and reside at the midline [35]. Some studies have reported that open transcranial approaches for lesions such as tuberculum sellae meningiomas result in higher rates of total resection and fewer complications such as cerebrospinal fluid (CSF) leak, while others suggest endonasal approaches may yield superior outcomes with respect to vision [27,36].

Ultimately, sellar lesions can be managed with different approaches [28,37]. Decision-making is nuanced and must be patient-specific; the optimal approach comes down to surgeon preference and experience. Familiarity with both transsphenoidal and transcranial techniques should be achieved so that both can be at the surgeon’s disposal.

### 4.1. Transsphenoidal Endoscopic Endonasal Approach

Transnasal transsphenoidal surgery is the most common approach to pituitary tumors that are confined to the sella and parasella [38]. Transsphenoidal surgery was performed and perfected under the microscope in the 20th century, but the use of the endoscope led to a boom in popularity while greatly enhancing the experience. Michael Apuzzo at the University of Southern California began to use the endoscope as an adjunct to conventional transsphenoidal surgery in 1977 [15,39], but it was not until the 1990s that purely endoscopic endonasal transsphenoidal surgery to the pituitary gained a true foothold, and Ricardo Carrau and Hae-Dong Jho published the first large case series of their experience with the endoscope, operating on 50 patients [17,40]. Today, endoscopic transsphenoidal surgery is a preferred route for pituitary adenomas and many sellar tumors because of the higher likelihood of preserving pituitary and visual function [31,41].

The endoscopic endonasal procedure can be performed independently or by two surgeons. A multidisciplinary effort may include an otolaryngologist driving the endoscope and establishing the access corridor to the sphenoid sinus, followed by a neurosurgeon dissecting the sinus and the tumor. The decision to collaborate with otolaryngology or operate as a single surgeon varies based on institution and surgeon preference. During the approach, key landmarks including the vomer, sellar floor, sphenoid rostrum, and opticocarotid recesses should be identified [7]. The procedure has been well described in the literature [42]. In brief, the endoscope is introduced to the nasal cavity and navigated to the sphenoid ostium. A bilateral sphenoidotomy and posterior nasal septectomy expose the keel of the sphenoid rostrum; this facilitates opening of the sellar floor and observation of the dura [1]. During pituitary tumor resection, the lateral boundaries are the cavernous sinuses bilaterally. Extension of the approach to access parasellar and clival tumors can be achieved by repositioning self-retaining retractors and resecting additional bone in the superior, inferior, and lateral directions [18,43]. The main advantages of the endoscopic endonasal approach are minimized brain retraction and superior visualization and illumination of the sellar anatomy [19,41]. The additional use of intraoperative navigation with stereotactic CT imaging can further reduce operative morbidity without an increase in operative time [44].

The approach is versatile, but it is mainly used for midline tumors. Access to compartments laterally beyond the optic nerves, cranial nerves, and surrounding vasculature can be limited [14,31]. There is a risk of cerebrospinal fluid leak—reports state it is as high as 30%, and a cerebrospinal fluid fistula can occur [13,28]. The two-dimensional field of view that is offered by endoscopic navigation can be a limitation, as can the restricted surgical instrument maneuverability compared with open surgery and a possibly significant learning curve required to master the endoscope [41]. Neurosurgeons reach proficiency operating under the binocular microscope during training and throughout their careers, but endoscopic surgery introduces a new set of instruments that require familiarization and skills that must be sharpened to achieve satisfactory results [45]. When resecting tumors near the midline, inadvertent injury to the internal carotid arteries, cavernous sinus, or optic nerves is possible, so spatial awareness and careful dissection technique is essential [7]. Transcranial approaches offer surgical maneuverability and can effectively remove tumors of the sella/parasella, but increasing application of minimally invasive endoscopic approaches has been shown to speed recovery, reduce perioperative risks, and lead to comparable or, in some cases, improved outcomes [46].

### 4.2. Pterional Approach

The pterional approach is the anterolateral workhorse in the skull base neurosurgeon’s armamentarium (Figure 6). It was described by Walter Dandy in 1938 and popularized by Gazi Yaşargil [32]. It is centered on the sylvian fissure over the sphenoid ridge and provides an operative window to the suprasellar, paraclinoid, and parachiasmal spaces [1,47]. The main advantage to the pterional approach is its wide access and ability to gain early vascular control, but the procedure is maximally invasive, can lead to postoperative cosmetic defects from the craniotomy, and frequently results in atrophy of the facial nerve and temporalis muscle [32]. Modifications to the pterional craniotomy include minimally invasive options such as the lateral supraorbital and mini-pterional approaches.

### 4.3. Orbitozygomatic Approach

The orbitozygomatic approach is as maximally invasive as the pterional approach. It is essentially a modified pterional craniotomy with additional removal of the superior and lateral orbital rim plus a zygomatic osteotomy [1]. It can be advantageous when treating challenging lesions in the parasellar region, interpeduncular fossa, and tumors with significant superior and lateral extension from the sella turcica such as tuberculum sellae meningiomas [48]. The additional removal of the zygoma, superolateral orbital rim, and orbital wall facilitates improved multidirectional operative corridor access, surgical maneuverability, and reduced brain retraction at the expense of greater bone removal [47]. The disadvantages and risk profile are similar to those of the pterional approach.

### 4.4. Mini-Pterional Approach

The mini-pterional approach uses a smaller craniotomy than a pterional approach, with a transsylvian dissection that yields an exposure comparable with that of the standard procedure [47,49]. Centered on the sphenoid ridge, an incision is made approximately 1 cm above the zygomatic arch at the anterior border of the hairline and extended to the ipsilateral midpupillary line. A burr hole is made superior to the frontozygomatic suture, beneath the line temporalis, and an osteotomy is made elevating a bone flap that includes part of the frontal bone inferior to the superior temporal line, a minimal portion of the temporal bone, and the lateral aspect of the sphenoid bone [49]. The reduced size of the craniotomy lessens temporalis muscle dissection and tissue trauma, minimizes lobar retraction, and improves cosmetic outcomes without sacrificing the working angles of the approach [49]. The operative corridor is inherently more limited than the pterional approach, thus the mini-pterional is a suitable alternative in properly selected patients [47].

### 4.5. Lateral Supraorbital Approach

An additional minimally invasive modification of the pterional approach is the lateral supraorbital craniotomy (Figure 7) [32,50]. Using an incision behind the hairline, above the pinna, and extending medially towards the midpoint between the superior temporal line and the midline, the lateral supraorbital approach dissects a small portion of the superior and anterior temporalis muscles and requires a single, small burr hole at the MacCarty keyhole.

A small bone flap is elevated with the keyhole at the center of the inferior margin [32]. The minimally invasive exposure is comparable with that of the pterional approach to reach parachiasmal lesions with a direct route [32,51]. In properly selected patients, the lateral supraorbital approach can be used as an alternative to the pterional approach for tuberculum sellae and planum sphenoidale meningiomas [32].

### 4.6. Supraorbital Approach

A supraorbital craniotomy can access deep-seated sellar/parasellar lesions of the tuberculum sellae, planum sphenoidale, and anterior clinoid process with minimal brain retraction [1,45,52]. It was first described by John Jane, Sr. in 1982 [47]; it is a suitable keyhole approach to the anterolateral skull base [53]. A short skin incision within the eyebrow is made along the superior orbital rim, avoiding the supraorbital nerve medially and the frontal branch of the facial nerve superiorly, and a superior orbital rim osteotomy is performed [1,28,47,52]. The limited skin incision yields a good cosmetic outcome, and attention is paid to preserve the frontal branch of the facial nerve and superficial temporal artery [52]. Like other keyhole approaches, limitations to the supraorbital craniotomy include narrow viewing angles, smaller working corridor, and decreased exposure width [45].

### 4.7. Interhemispheric and Combined Endoscopic Approaches

There are a multitude of other approaches to reach the sellar/parasellar region. Preclinical studies in cadaveric models have given rise to attempts at developing more direct, minimally invasive routes that retain the benefits of standard approaches. An anterior interhemispheric route favored by Bruneau and colleagues uses neuronavigation and a limited bicoronal incision behind the hairline with a midline anterior burr hole above the superior longitudinal sinus to reach tuberculum sellae meningiomas [12]. They describe their experience of achieving a symmetrical superior view of bilateral optic canals, optic nerves, and the optic chiasm with a unilateral incision while maintaining full proximal and distal control over vessels and visualizing posteroinferior tumor extension. An endoscopic intradural subtemporal keyhole approach as described by Ding et al. [54] uses frameless navigation and endoscope to reach the suprasellar, paraclival, and ventrolateral brainstem regions with minimal invasiveness. Additionally, a microscopic endoscope-assisted transmaxillosphenoidal approach evaluated by Gagliardi et al. [30] capably accessed the sellar, suprasellar, and parasellar region with minimal invasiveness, wide exposure, and avoidance of critical neurovascular structures encountered transsphenoidally by obtaining access through the maxillary sinus. The combined transmaxillary–transsphenoidal approach avoids violating the nasal cavity and obviates the need for a craniotomy [55]. Proper selection of patients is imperative during preoperative planning so that a surgical approach can be tailored on an individual basis.

## 5. Future Directions

Although the mainstays of treatment are microscopic, endoscopic, and keyhole transcranial and transsphenoidal approaches, innovations to enhance skull base surgery of the sellar/parasellar region are taking place. The evolution of three-dimensional technology to improve upon the two-dimensional visualization provided by endoscopes has occurred in the past decade [15]. The newly introduced exoscope, a high-definition telescopic device, has yet to be studied extensively in skull base surgery to understand its range of functionality [17]. Intraoperative use of agents such as indocyanine green, fluorescein, and 5-aminolevulinic acid are being implemented in endoscopic skull base surgery to improve safety and visualization. Fluorescein has been studied in pituitary adenomas and can differentiate tumor from healthy tissue, while indocyanine green and 5-aminolevulinic acid can enhance tumor dissection by highlighting nearby vessels and delineating the tumor as it is being dissected in difficult-to-visualize locations such as in proximity to the optic canals and cranial nerves [5,17]. Finally, there have been great strides in robotics in surgery. Although these applications are more often seen in other disciplines such as general surgery and orthopedics, robotics continues to be incorporated at a greater rate in neurosurgery. In skull base surgery, the use of robotic endoscope holders has been implemented with considerable success. Transoral robotic surgery has also been attempted to access the sella [41,56,57]. However, the large start-up costs to the hospital and the patient currently make robotic skull base surgery unfavorable [41]. In addition, with the current generation of robotics that were not specifically engineered for neurosurgical procedures, there are inherent limitations in maneuverability that do not permit access to the small, deep, working corridors of the skull base. Nor do the robots possess the fine, slight, dexterous movements necessary for the region [41,58]. Additional limitations that have been described include the lack of haptic feedback [17,57]. Nevertheless, as robotics continue to improve, their use in skull base neurosurgery is expected to expand. Their greatest utility will not come from replacing the surgeon’s skills but rather by improving intraoperative comfort and enhancing safety.

## 6. Conclusions

Surgery of the sellar and parasellar region is richly steeped in neurosurgical history. Through collaborative efforts among many of the greats and pioneers of the field, stepwise innovations and modifications have led to the development of the techniques that are regularly used today. A transsphenoidal approach to the sella, nearly abandoned at the midpoint of the 20th century, is the mainstay approach used in modern practice. Introduction of the operative microscope and then the endoscope brought forth a trove of new possibilities and expanded operative versatility for treating sellar/parasellar lesions. Future directions will be similarly established by boundary-pushing surgeons working towards a goal of improving and providing optimal treatment to patients.

## Figures and Tables

**Figure 1 cancers-15-02896-f001:**
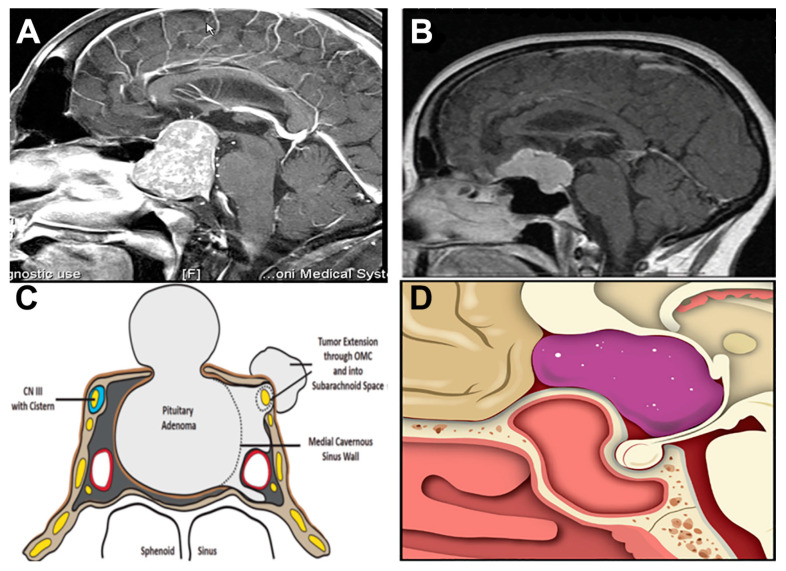
Neuroanatomy of the most common sellar lesions. (**A**,**C**) Pituitary adenomas originate from the gland as it sits in the sella turcica. Macroadenomas can extend to the parasellar, retrosellar, suprasellar, and diaphragma sellar space, which results in compression of neurological structures. Panel C reprinted from: Hoang, N., Tran, D. K., Herde, R., Couldwell, G. C., Osborn, A. G., and Couldwell, W. T. (2016). Pituitary macroadenomas with oculomotor cistern extension and tracking: implications for surgical management, *J. Neurosurg*. 125(2), 315–322. (**B**,**D**) Tuberculum sellae meningiomas may extend to the planum sphenoidale, clivus, and sinus wall. Panels B and D reprinted from Raheja, A., Karsy, M., Eli, I., Guan, J., and Couldwell, W.T. (2017). Endonasal operative corridor expansion by sphenoidal pneumosinus dilatans in tuberculum sellae meningiomas, *World Neurosurg.* 106, 686–692, with permission from Elsevier.

**Figure 2 cancers-15-02896-f002:**
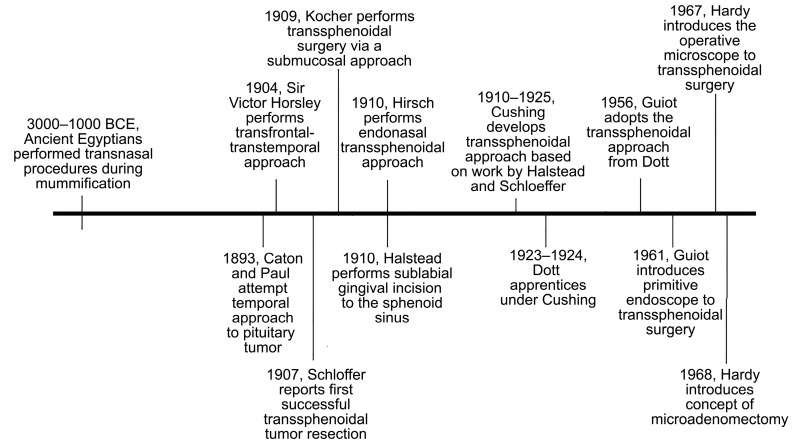
Timeline of events leading to development of transsphenoidal approaches to the sella.

**Figure 3 cancers-15-02896-f003:**
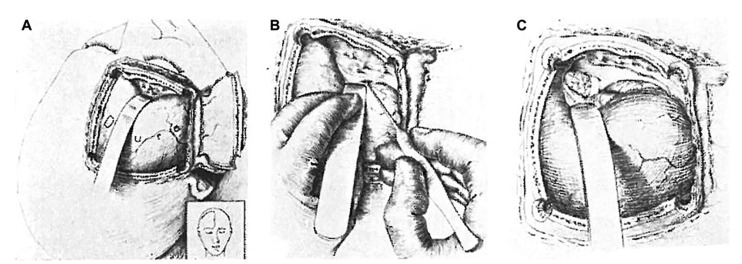
The transfrontal approach to the sella was favored by Cushing. His preference for transcranial surgery had great influence over the neurosurgical community in North America. Illustration of the (**A**) craniotomy, (**B**) retraction of the frontal dura, and (**C**) exposure of the tumor. Reprinted from Cushing H, Eisenhardt L. Meningiomas arising from the tuberculum sellae. With the syndrome of primary optic atrophy and bitemporal field defects combined with a normal sella turcica in a middle-aged person. *Arch. Ophthal*. 1929;1(2):168–206.

**Figure 4 cancers-15-02896-f004:**
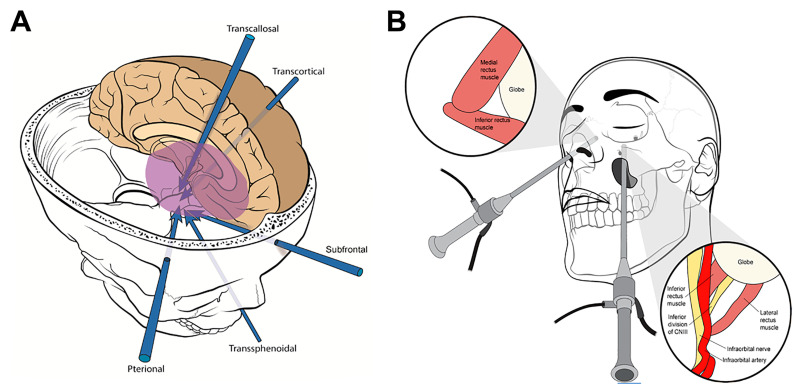
(**A**) Surgical approaches and trajectories to the sellar/parasellar region. Reproduced with permission of the Department of Neurosurgery, University of Utah. (**B**) Endoscopic endonasal approach and transantral approach with endoscopic assistance. © 2020 Abou-Al-Shaar, Krisht, Cohen, Abunimer, Neil, Karsy, Alzhrani, and Couldwell. Reproduced under the terms of the Creative Commons Attribution License (CC BY).

**Figure 5 cancers-15-02896-f005:**
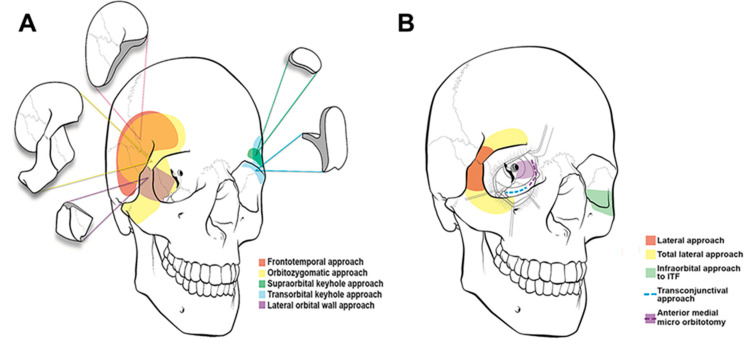
(**A**) Cranio-orbital and (**B**) orbito-cranial approaches to lesions in the anterior and middle cranial fossa. There is no one-size-fits-all approach. Selection of traditional open approaches versus minimally invasive and keyhole surgery is case-dependent. © 2020 Abou-Al-Shaar, Krisht, Cohen, Abunimer, Neil, Karsy, Alzhrani and Couldwell. Reproduced under the terms of the Creative Commons Attribution License (CC BY).

**Figure 6 cancers-15-02896-f006:**
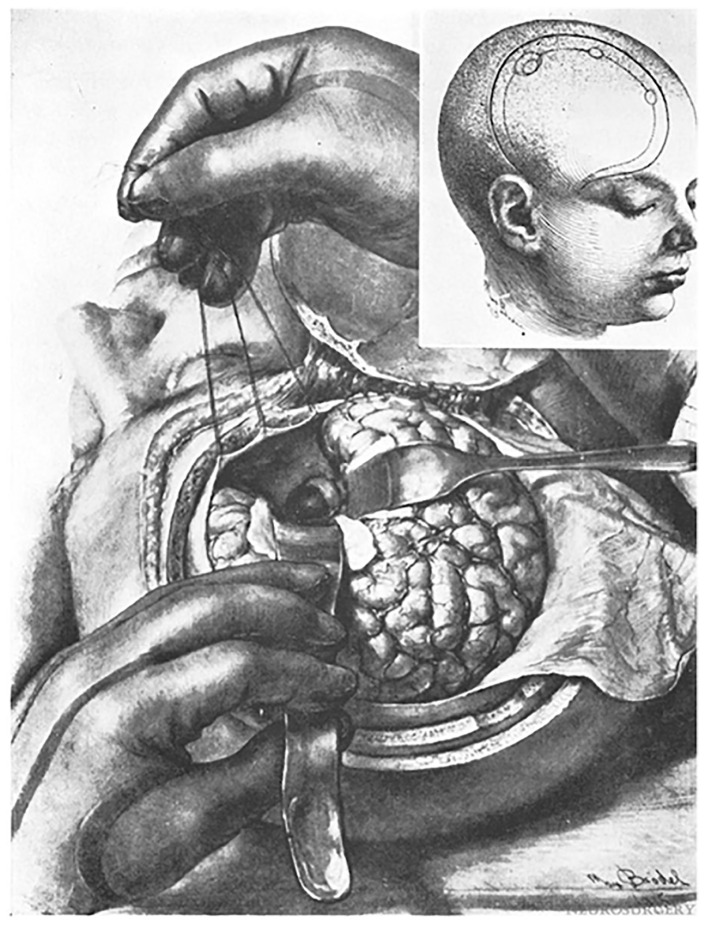
Illustration of an early rendition of a frontotemporal approach favored by Heuer. Reprinted from Heuer GJ. Surgical experiences with an intracranial approach to chiasmal lesions. *Arch. Surg*. 1920;1(1):368–381.

**Figure 7 cancers-15-02896-f007:**
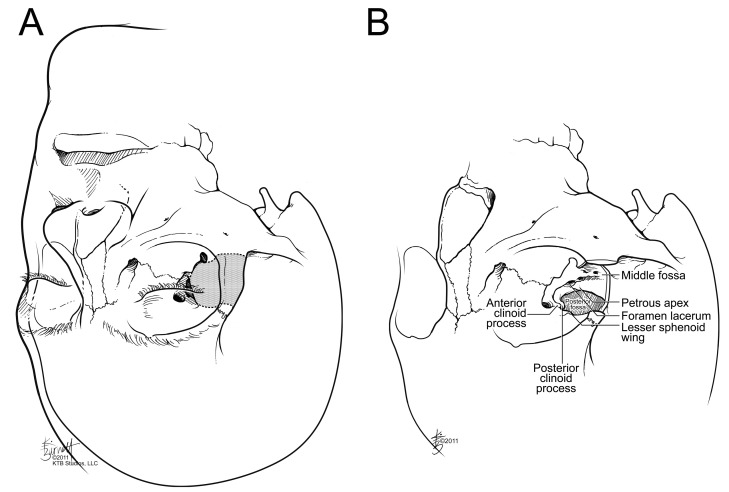
(**A**) Sellar/parasellar region accessible from a lateral orbital approach. (**B**) Illustration showing the parasellar area and middle fossa structures after removal of the lateral orbital rim and wall in the translateral orbital approach. (Reprinted with permission from KTB Studios, LLC.).

**Table 1 cancers-15-02896-t001:** Common surgical approaches to the sella/parasella.

Approach	Advantages	Disadvantages
Pterional	Wide angle of exposure, maximal surgical maneuverability, familiarity with approach. Can achieve early vascular control. Versatility, i.e., can manage large, invasive, complex neoplastic and vascular lesions.	Maximally invasive, extensive craniotomies. Longer surgery, hospital stay, and recovery time. Brain retraction injury. Risk of temporalis muscle atrophy and injury to the frontal branch of facial nerve.
Frontolateral		
Orbitozygomatic		
Fronto-orbitozygomatic		
Bifrontal craniotomy		
Transsphenoidal	Direct, midline approach to the sella. Versatile, can be performed with microscopic or endoscopic assistance. Superior visual outcomes, no extrinsic cosmetic defects.	Limited to midline and paramedian lesions. Lateral extension/invasion may require a more open approach. Endoscopic surgery requires additional training and specialized equipment. High risk of CSF leak.
Lateral supraorbital	Keyhole and minimally invasive approaches. Direct route to lesions. Less blood loss, brain retraction. Lower risk of cosmetic defect.	Requires proper patient selection. Worse surgical maneuverability and more limited operative corridor compared with traditional open surgery. More challenging management of perioperative complications.
Supraorbital eyebrow approach		
Lateral orbitotomy		
Supraorbital		
Mini-pterional		
Anterior interhemispheric		

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
