# Peer review of "History, Current Techniques, and Future Prospects of Surgery to the Sellar and Parasellar Region"

_cancers, 2023, doi:10.3390/cancers15112896_

Round 1

Reviewer 1 Report

Sellar and parasellar surgery is a technical challenge because of precious anatomical structures in this region. Several techniques have been developed in the past 150 years, either transcranial or transsphenoidal. The authors propose to review this surgery, based on historical, anatomical and technical considerations.

Globally, this article is a subjective discussion of different techniques, with historical considerations. At the end of it, it seems that all techniques are equal, with pros and cons, which does not reflect the scientific reality, with some tumors associated with documented best approaches (for instance adenomas/transsphenoidal whereas meningiomas/transcranial). Since it is not a systematic review and does not offer original illustrations (that could be useful to non neurosurgeons) or technical suggestions or guidelines, we found that the most interesting part is the historical paragraph about the re-birth of transsphenoidal surgery. We suggest that this article be reorganized as a historical article.

Comments:

-          “In the modern era, transcranial and transsphenoidal approaches to the sel- 29 lar and parasellar region, with endoscopic or microscopic technique, are regularly used 30 without any clear distinction in superiority; both possess advantages and limitations” This assumption in the introduction is already very controversial and vague. It needs to be removed or much more detailed. Indeed, both techniques are used in particular cases, and each has proved superior in some cases. This sentence could make believe that the two techniques are always possible and equal.

-          The “Anatomy” part and figure are purely descriptive and the challenges associated with each type of tumor is not explicated, as would be necessary for non neurosurgeons/less experienced neurosurgeons. The therapeutic implications of diagnosing a meningioma rather than a pituitary adenoma are massive in terms of surgical decisions and techniques.

-          Also, considering the impact of the journal, and the large targeted medical audience, the authors would improve the review by creating original pictures to illustrate their purpose (all figures).

-          History part. “The earliest accounts of surgical approaches to the sella can be credited to the ancient 70 Egyptians[16] (Figure 2).” Although the Egyptian reference is exciting, it does not seem to be relevant in case of sellar surgery since the Egyptians used it to reach the brain. Some explanation is needed for non neurosurgeons, and to explain why they did not perform sellar surgery at all (cf. anatomical explanation).

-          Some anatomical description of what trans-sphenoidal or transglabellar approaches are is needed, through a figure for instance.

-          Figure 2 is relatively poor compared to reading the text itself.

-          “The survival, and eventual revival, of transsphenoidal surgery is largely credited to 131 the work of Norman Dott, Gerard Guiot, and Jules Hardy. Dott, as a fellow from Edin- 132 burgh, observed the transsphenoidal approach to the pituitary from Cushing and adopted 133 it into his own practice [18].” This is by far the most interesting and richly documented item of the historical part. We could suggest that this becomes the central idea of the article.

-          Current techniques part : “The decision to approach a 167 tumor via open approach versus endoscopically is based on surgeon preference; often, 168 both methods are viable options for the same skull-base lesion”. Once again, this is a very general consideration and gives a simplistic idea of what approaches are. We suggest that this is more detailed with pros and cons explicited and documented with literature references. “Familiarity with both transsphenoidal and transcranial techniques 205 should be achieved so that both can be at the surgeon’s disposal. » We therefore completely agree with this conclusion.

-          Table 1 seems a very subjective list of advantages and disadvantages. Literature references may be made clearer. For instance there is no mention of brain damade as a disadvantage of transcranial surgery, whereas it is mentioned in the text above.

-          Transphenoidal approach. This part is too long and many details are based on the authors’ personal experience rather than a review perspective. For instance, the simple sentence “The endoscopic endonasal procedure requires two surgeons.” Is not true in my experience, since endoscopic surgery can be performed with a single surgeon, either holding the endoscope in one hand and the instruments in one or both hands, or using a surgical arm to hold the microscope and work with both hands.

-          Transcranial approaches need to be illustrated appropriately. Especially the minipterional one.

Author Response

Reviewer #1:

Sellar and parasellar surgery is a technical challenge because of precious anatomical structures in this region. Several techniques have been developed in the past 150 years, either transcranial or transsphenoidal. The authors propose to review this surgery, based on historical, anatomical, and technical considerations. Globally, this article is a subjective discussion of different techniques, with historical considerations. At the end of it, it seems that all techniques are equal, with pros and cons, which does not reflect the scientific reality, with some tumors associated with documented best approaches (for instance adenomas/transsphenoidal whereas meningiomas/transcranial). Since it is not a systematic review and does not offer original illustrations (that could be useful to non-neurosurgeons) or technical suggestions or guidelines, we found that the most interesting part is the historical paragraph about the re-birth of transsphenoidal surgery. We suggest that this article be reorganized as a historical article.

  1. In the modern era, transcranial and transsphenoidal approaches to the sel- 29 lar and parasellar region, with endoscopic or microscopic technique, are regularly used 30 without any clear distinction in superiority; both possess advantages and limitations” This assumption in the introduction is already very controversial and vague. It needs to be removed or much more detailed. Indeed, both techniques are used in particular cases, and each has proved superior in some cases. This sentence could make believe that the two techniques are always possible and equal.

RESPONSE: We have revised the language in the text so that this statement is less matter of fact and instead is solely a comment on approaches to the sellar region. Page 1; lines 21-32.

  1. The “Anatomy” part and figure are purely descriptive, and the challenges associated with each type of tumor is not explicated, as would be necessary for non-neurosurgeons/less experienced neurosurgeons. The therapeutic implications of diagnosing a meningioma rather than a pituitary adenoma are massive in terms of surgical decisions and techniques.

RESPONSE: The intent of this section was to be descriptive with supportive figures illustrating examples of the most common pathologies. We consider the challenges associated with diagnosing and treating each type of tumor to be beyond the scope of this article as a whole – the frequency of pituitary adenomas affecting the sellar region and the development and evolution of surgery to manage these lesions is, in our opinion, the importance of briefly introducing the anatomy.

  1. -          Also, considering the impact of the journal, and the large targeted medical audience, the authors would improve the review by creating original pictures to illustrate their purpose (all figures).

RESPONSE: We appreciate the reviewer’s comment. Many of the pictures are original from our institution and accurately portray the anatomy, current operative approaches, and historical techniques. We believe these examples adequately support the text.

  1. History part. “The earliest accounts of surgical approaches to the sella can be credited to the ancient 70 Egyptians[16] (Figure 2).” Although the Egyptian reference is exciting, it does not seem to be relevant in case of sellar surgery since the Egyptians used it to reach the brain. Some explanation is needed for non-neurosurgeons, and to explain why they did not perform sellar surgery at all (cf. anatomical explanation).

RESPONSE: We mention the Egyptians to depict how early, primitive transnasal routes existed centuries before formal neurosurgery procedures. We have edited the text to emphasize that these practices were not surgical in nature – they were for cultural purposes. Page 2; Lines 73-77.

  1. Some anatomical description of what trans-sphenoidal or transglabellar approaches are is needed, through a figure for instance.

RESPONSE: A description of the transsphenoidal approach is provided in section 4.1. Figure 4A,B displays the trajectory of the transsphenoidal approach.

  1. Figure 2 is relatively poor compared to reading the text itself.

RESPONSE: This figure was provided as an additional visual aid to outline the historical timeline concisely. If the reviewers prefer, we can remove this figure from the article, but we thought to keep it for visual interest.

  1. “The survival, and eventual revival, of transsphenoidal surgery is largely credited to 131 the work of Norman Dott, Gerard Guiot, and Jules Hardy. Dott, as a fellow from Edin- 132 burgh, observed the transsphenoidal approach to the pituitary from Cushing and adopted 133 it into his own practice [18].” This is by far the most interesting and richly documented item of the historical part. We could suggest that this becomes the central idea of the article.

RESPONSE: Thank you for your comment. We agree that the history and revival of transsphenoidal surgery is profound and of interest to broader readership; however, we have written previously on this subject. We refer the reviewer particularly to reference 21, “Norman Dott, Gerard Guiot, and Jules Hardy: key players in the resurrection and preservation of transsphenoidal surgery.” We believe this section is central to the article but should remain in its current location.

  1. Current techniques part : “The decision to approach a 167 tumor via open approach versus endoscopically is based on surgeon preference; often, 168 both methods are viable options for the same skull-base lesion”. Once again, this is a very general consideration and gives a simplistic idea of what approaches are. We suggest that this is more detailed with pros and cons explicated and documented with literature references. “Familiarity with both transsphenoidal and transcranial techniques 205 should be achieved so that both can be at the surgeon’s disposal. » We therefore completely agree with this conclusion.

RESPONSE: Thank you for bringing this to attention. We agree that the lines 167-168 mentioned above are general. Previously: Pages 5-6, lines 188-196 included some of the pros and cons of different approaches. Moreover, operative planning and surgical decision making is an extensive topic that can have its own dedicated review, although we believe that is beyond the scope of the current submission. We have added new references and further explained how an open versus endoscopic approach can be selected. Now: Pages 6-7; lines 213-231.

  1. Table 1 seems a very subjective list of advantages and disadvantages. Literature references may be made clearer. For instance there is no mention of brain damade as a disadvantage of transcranial surgery, whereas it is mentioned in the text above.

RESPONSE: Thank you for raising this point. We have revised the second column to reference brain damage as a possible risk of transcranial surgery. Table 1 was intended to be a concise summary of some of the pros and cons of the surgical approaches covered in the text. If the reviewers deem this table unnecessary, we can remove it from the article.

  1. Transphenoidal approach. This part is too long and many details are based on the authors’ personal experience rather than a review perspective. For instance, the simple sentence “The endoscopic endonasal procedure requires two surgeons.” Is not true in my experience, since endoscopic surgery can be performed with a single surgeon, either holding the endoscope in one hand and the instruments in one or both hands, or using a surgical arm to hold the microscope and work with both hands.

RESPONSE: In our reply above to point #5, we included a longer description of the transsphenoidal approach to describe the anatomy, approach, and practical uses. The approach is a mainstay in sellar/parasellar surgery because of the frequency with which pituitary adenomas are diagnosed relative to other regional pathology. We have revised this section and the language to remove some of the subjectivity and provide more of a review perspective. (Page 7, lines 249-253.)

  1. Transcranial approaches need to be illustrated appropriately. Especially the minipterional one.

RESPONSE: The use of the previously published illustrations was intended to provide a near-comprehensive overview of the many approaches available to the skull-based neurosurgeon. The operative trajectories for transcranial approaches are depicted in Figure 4a and further supported by the cranitome illustrations in Figures 5a and 5b. Figure 3 and Figure 6 illustrate the early renditions of transcranial surgery practiced by Cushing and Heuer, respectively.

Reviewer 2 Report

The authors present a well-reviewed study of the history, current techniques, and future prospects of surgery to the sellar and parasellar region. It is a well studied compilation. 

Author Response

Reviewer #2:

The authors present a well-reviewed study of the history, current techniques, and future prospects of surgery to the sellar and parasellar region. It is a well studied compilation. 

RESPONSE: Thank you.

Reviewer 3 Report

The authors revied the history, current techniques, and future prospects of surgical approach in patients with sellar and parasellar disease. They fully reviewed the theme of the title.

Author Response

Reviewer #3:

The authors revied the history, current techniques, and future prospects of surgical approach in patients with sellar and parasellar disease. They fully reviewed the theme of the title.

RESPONSE: Thank you.